

# Changes in zooplankton community, and seston and zooplankton fatty acid profiles at the freshwater/saltwater interface of the Chowan River, North Carolina

Deborah A. Lichti[1], Jacques Rinchard[2] and David G. Kimmel[1,3]

[1] Department of Biology, East Carolina University, Greenville, NC, United States of America
[2] Department of Environmental Science and Biology, State University of New York College at Brockport, Brockport, NY, United States of America
[3] Alaska Fisheries Science Center, National Oceanic and Atmospheric Administration, Seattle, WA, United States of America

Corresponding author
Deborah A. Lichti,
lichtid12@students.ecu.edu

## ABSTRACT

The variability in zooplankton fatty acid composition may be an indicator of larval fish habitat quality as fatty acids are linked to fish larval growth and survival. We sampled an anadromous fish nursery, the Chowan River, during spring of 2013 in order to determine how the seston fatty acid composition varied in comparison with the zooplankton community composition and fatty acid composition during the period of anadromous larval fish residency. The seston fatty acid profiles showed no distinct pattern in relation to sampling time or location. The mesozooplankton community composition varied spatially and the fatty acid profiles were typical of freshwater species in April. The Chowan River experienced a saltwater intrusion event during May, which resulted in brackish water species dominating the zooplankton community and the fatty acid profile showed an increase in polyunsaturated fatty acids (PUFA), in particular eicosapentaenoic acid (EPA) and docosahexaenoic acid (DHA). The saltwater intrusion event was followed by an influx of freshwater due to high precipitation levels in June. The zooplankton community composition once again became dominated by freshwater species and the fatty acid profiles shifted to reflect this change; however, EPA levels remained high, particularly in the lower river. We found correlations between the seston, microzooplankton and mesozooplankton fatty acid compositions. Salinity was the main factor correlated to the observed pattern in species composition, and fatty acid changes in the mesozooplankton. These data suggest that anadromous fish nursery habitat likely experiences considerable spatial variability in fatty acid profiles of zooplankton prey and that are correlated to seston community composition and hydrodynamic changes. Our results also suggest that sufficient prey density as well as a diverse fatty acid composition is present in the Chowan River to support larval fish production.

# INTRODUCTION

Estuaries are considered important nursery habitat for many ecologically and commercially important fish and invertebrates (*Beck et al., 2001*; *Boesch & Turner, 1984*; *Sheaves et al., 2015*; *Sheaves, 2016*). Estuaries function as fish nurseries because they are highly productive, support large planktonic populations across multiple size ranges, and fish within estuaries generally have higher growth rates compared to other habitats (*Beck et al., 2001*). Hence, many fish have evolved life-history strategies whereby larvae and juvenile stages have residency periods in estuaries (*McHugh, 1967*; *Boehlert & Mundy, 1988*; *Beck et al., 2001*; *Able, 2005*; *Walsh, Settle & Peters, 2005*). *Beck et al. (2001)* defined nursery habitat as an area that contributes to higher production of individuals that move to juvenile habitat compared to other areas. Higher growth rates of larval fish are possible because of zooplankton prey that are present during their critical transition from yolk sac to free-living, feeding larvae (*Hjort, 1914*; *Mullen, Fay & Moring, 1986*; *Rulifson et al., 1993*; *Cooper et al., 1998*; *Martino & Houde, 2010*; *Binion, 2011*). However, spatial and temporal overlap between predators and prey does not completely explain how fish nurseries function mechanistically. The quality of prey can play a major role in determining the effectiveness of a nursery for early stages of fish.

The quality (chemical composition) of zooplankton prey can influence fish growth, development, and survival (*Fraser et al., 1989*; *Webster & Lovell, 1990*; *Copeman et al., 2002*; *Rossi et al., 2006*; *Malzahn et al., 2007*; *Paulsen et al., 2014*). Fatty acids are chemically diverse, often incorporated into organisms unmodified, and different organisms have distinct profiles (*Dalsgaard et al., 2003*). Fatty acids are one class of compounds found in lipids that are particularly important, impacting neural and vision development in fish (*Gulati & Demott, 1997*; *Müller-Navarra et al., 2000*; *Kainz, Arts & Mazumder, 2004*; *Masclaux et al., 2012*). Fatty acids may act as both dietary tracers in the food web and indicators of overall food quality (*Iverson et al., 2004*). The majority of organisms need specific dietary fatty acids for somatic development and fitness (*Masclaux et al., 2012*). These fatty acids, 18:3ω-3, α-linolenic acid (ALA), and 18:2ω-6, linoleic acid (LA), are labeled essential fatty acids because they cannot be directly synthesized by heterotrophic organisms and must come from the diet (*Arts, Brett & Kainz, 2009*). Polyunsaturated fatty acids (i.e., 20:5ω-3, eicosapentaenoic acid (EPA), 22:6ω-3, docosahexaenoic acid (DHA), and 20:4ω-6, arachidonic acid (ARA)) are required for all organisms and play a role in health and cell function (*Dalsgaard et al., 2003*). Thus, an organisms' fatty acid signature may indicate dietary consumption and nutritional quality of its prey (*Goncalves et al., 2012*).

Fatty acids are present in estuaries as a result of *de novo* synthesis by phytoplankton and the delivery of detrital material of plant origin (*Dalsgaard et al., 2003*). The free-floating portion of organic and inorganic particles is termed seston (*Postel, Fock & Hagen, 2000*). The organic portion of seston is important because it forms the origin point for the propagation of fatty acids through the pelagic food web. Zooplankton assimilate fatty acids from the seston through direct consumption of phytoplankton cells, detritus and/or consumption of microzooplankton that graze phytoplankton or detritus (*Wacker & Von Elert, 2001*; *Kainz, Arts & Mazumder, 2004*; *Vargas, Escribano & Poulet, 2006*). To date,

studies of the relationship between seston and zooplankton fatty acid composition have shown variable patterns. *Persson & Vrede (2006)* demonstrated in laboratory studies that broad zooplankton taxonomic groups (cladoceran vs. copepods) have different fatty acid profiles independent of the food source. The seston has been correlated to the fatty acid profile of zooplankton *in situ* as well (*Goulden & Place, 1990*; *Brett et al., 2006*; *Taipale et al., 2009*; *Gladyshev et al., 2010*; *Ravet, Brett & Arhonditsis, 2010*). The mismatch of fatty acids in seston to zooplankton has also been shown in many studies (*Desvilettes et al., 1997*; *Persson & Vrede, 2006*; *Rossi et al., 2006*; *Smyntek et al., 2008*). Most marine zooplankton species cannot convert precursor fatty acids and only obtain longer-chained fatty acids from their diet, compared to freshwater zooplankton species that can convert precursor fatty acids (*Rossi et al., 2006*; *Persson & Vrede, 2006*). However, it is clear that the fatty acid composition of the seston changes as a result of local conditions, e.g., the salinity and temperature, nutrient concentration, and the degree of autotrophy or heterotrophy in the system (*Farkas & Herodek, 1964*; *Desvilettes et al., 1997*; *Wacker & Von Elert, 2001*; *Goncalves et al., 2012*). Therefore, combined knowledge of the changing nature of seston fatty acid composition, zooplankton community composition changes, and fatty acid profiles forms a useful base for assessing the quality of fish nursery habitat.

Zooplankton community composition in estuaries has been intensely studied and abiotic factors are thought to structure zooplankton communities (*Ambler, Cloern & Hutchinson, 1985*; *Orsi & Mecum, 1986*; *Cervetto, Gaudy & Pagano, 1999*; *Mouny & Dauvin, 2002*; *Kimmel & Roman, 2004*; *Lawrence, Valiela & Tomasky, 2004*; *Islam, Ueda & Tanaka, 2005*). Zooplankton community composition in temperate estuaries is dominated by crustaceans in general, and copepods and cladocerans in particular (*Tackx et al., 2004*; *Marques et al., 2006*; *Winder & Jassby, 2011*; *Chambord et al., 2016*). Cladocerans are characterized by high levels of EPA and ARA and this is thought to be related to a life history strategy focused on high rates of somatic growth (*Persson & Vrede, 2006*). In contrast, copepods have higher relative DHA levels because this fatty acid is critical for nervous system development (*Arts, Brett & Kainz, 2009*). Copepods feature more developed nervous systems compared to cladocerans and this is a function of active hunting of prey, mate location, and predator avoidance (*Dalsgaard et al., 2003*). Carnivorous crustacean zooplankton have shown to be richer in PUFAs and this is thought to be related to their food source (rotifers and smaller bodied cladoceran/copepods compared to phytoplankton) (*Arts, Brett & Kainz, 2009*).

Here we explore the species composition and variability in fatty acid composition of the lower food web at the freshwater/saltwater interface of an estuarine fish nursery, the Chowan River, North Carolina, USA. The Chowan River is considered a critical habitat for larval and juvenile blueback herring (*Alosa aestivalis*), alewife (*A. pseudoharengus*), collectively known as river herring (*NCDMF, 2007*). The river herring are of interest because they have been severely overfished and a moratorium on harvest is in place at various locations along the eastern United States, including North Carolina (*ASMFC, 2012*). The Chowan River also serves as a nursery habitat for American shad (*A. sapidissima*) and striped bass (*Morone saxatilis*); however, the status of the habitat for the latter species is unknown (*Greene et al., 2009*).

The overall goal of our study was to determine if species and fatty acid composition of the lower food web could be used to indicate habitat quality of an estuarine fish nursery. In order to achieve this goal, we examined the spatial and temporal variability in species composition of microzooplankton and mesozooplankton as well as the fatty acid composition of the seston, microzooplankton, and mesozooplankton during the period of larval fish residency in the Chowan River. Our specific objectives were to determine: (1) if differences in the species composition and fatty acid composition were present in the system; (2) if so, were there patterns in species composition and fatty acid composition in time and space; (3) if particular species were related to the species composition patterns and if particular fatty acids were related to the fatty acid composition patterns; (4) if changes in species and fatty acid composition were related to changes in salinity dynamics; (5) if patterns in fatty acid composition correlated across trophic levels. We hypothesized that species composition would be related to salinity and that fatty acid profiles of micro- and mesozooplankton would relate to species composition and would reflect that of the seston. If supported, this would suggest that the quality of the larval fish forage, based on fatty acids, could be used to assess fish nursery quality.

## MATERIALS AND METHODS

### Study site

The Chowan River is one of the largest tributaries that drains into the Albemarle Sound (Figs. 1A and 1B) and is the 12th largest river basin in North Carolina (*NCDENR, 2006*). It is mainly a freshwater estuary that experiences intermittent salinity intrusion, mainly in the winter months (*Leech, Ensign & Piehler, 2009*). The Chowan River was classified as "nutrient sensitive waters" in 1979 (*NCDENR, 2006*) and has routinely experienced algal blooms and low dissolved oxygen levels (<3.0 mg L$^{-1}$). The entire river is classified as a Strategic Habitat Area for larval and juvenile river herring (*NCDMF, 2007*). Sampling took place south of Holiday Island on a 34 km transect of Chowan River (Fig. 1C). Seven locations (4 km apart) were sampled between Holiday Island and the river mouth (Fig. 1C). Sampling occurred on 10–11 April, 31 May, and 25 June 2013, dates that span the residency for larvae of alewife, blueback herring, and striped bass. For our study, we divided the river into three sections: upper, middle, and lower. The main differences among these sections were (1) the distance from the Albemarle Sound and (2) the potential influence of salinity.

### Sample collection

*Water column properties.* Vertical profiles of temperature (°C) and salinity were measured with a conductivity, temperature, and depth sensor (CTD, Yellow Springs Instruments, Castaway). Water samples were collected at a depth of 3 m with a Niskin water sampler.

*Zooplankton.* Water depths ranged from 5.27 to 7.56 m during zooplankton sampling. Two horizontal net tows were made with 0.5 m diameter nets of two different mesh sizes (60 and 200 μm). Two mesh sizes were used in order to generate an adequate representation of the zooplankton for the size range >60 μm. The zooplankton samples between 60 and

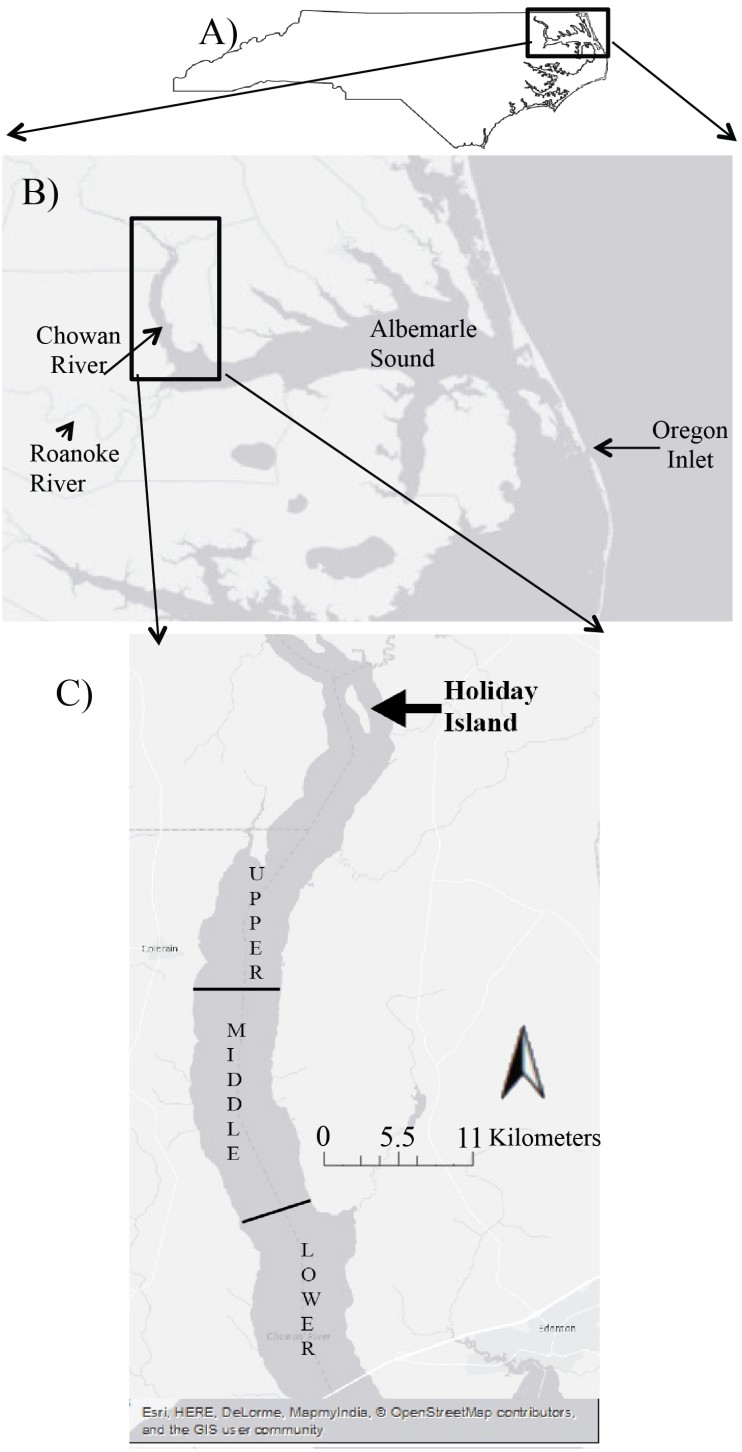

**Figure 1** **The overview of Albemarle Sound in North Carolina (A). The close up view of the location for two main tributaries (Chowan and Roanoke Rivers), and the Albemarle Sound in North Carolina (B). The three sections used to collect zooplankton on the Chowan River (C).** Map data: ESRI, HERE, DeLorme, MapmyIndia, OpenStreetMap contributors, GIS user community.

200 µm are designated microzooplankton and the >200 µm zooplankton samples are designated mesozooplankton throughout the remainder of the paper. The zooplankton net was towed obliquely through the water for 1 min (species composition) and 2 min (fatty acid composition) at an average boat speed of 0.75 m s$^{-1}$. The volume filtered was calculated using the volume of a cylinder ($V = \pi r^2 L$), where $r$ was the radius of the plankton net (0.25 m) and $L$ was determined using the boat speed (m s$^{-1}$) and the tow time (s). Each identification and count sample, depending on mesh size, was filtered through a 60 or 200 µm filter, and zooplankton for composition were preserved in a 120 mL glass jar with 10 ml of 10% buffered formaldehyde, sucrose, and filtered water. The addition of sucrose to the formalin helps to reduce ballooning of cladoceran bodies and inflation of their carapace (*Haney & Hall, 1973*). The 60 µm sample had a half tablet of Alka Seltzer added to keep rotifers from pulling in critical body parts (legs and arms) to ease identification (*Chick et al., 2010*). The zooplankton samples for fatty acid analysis were collected at seven sites on the Chowan River for April and June, and three sites in May. Due to limitations related to sampling preparation in May, a subset of the field sites was sampled for fatty acid analyses. The zooplankton samples for fatty acids were placed in a 1,000 mL plastic container on ice, and processed in the laboratory.

### Laboratory processing

*Zooplankton identification.* Samples were filtered through a sieve (60 or 200 µm) to remove the sugar formalin solution, and then added to a beaker with a known volume of water. A total of three subsamples (2 mL per subsample for microzooplankton and 5 mL per subsample for mesozooplankton) were analyzed for community composition using a Hensen-Stempel pipette. Organisms were identified using a dissecting microscope and enumerated using a Ward counting wheel. The zooplankton were identified to genus except for the freshwater copepods that were identified to order. Copepod nauplii were grouped together because identification can be difficult at this stage (*Johnson & Allen, 2012*). If a species in a subsample comprised greater than 500 individuals, then that species was not counted for the other two subsamples. Species abundances ($A$) were determined using the equation: $A = A_s(V_t V_s^{-1})$ where $A_s$ is the number of individuals in the subsample, $V_t$ is the total volume of water in the beaker, and $V_s$ is the volume of the subsample.

*Lipid and fatty acid samples.* The water samples (300 mL) were concentrated on a 0.7 µm Whatman$^{TM}$ GF/F filter (47 mm diameter) and stored at −80 °C until ready to process, which constituted the seston material. The zooplankton samples were filtered through 60 and 200 µm sieves stacked to collect species based on size. Each sample was visually analyzed to determine the dominant species with a dissecting microscope, and detritus and phytoplankton were removed. The samples were concentrated on a GF/F filter (47 mm diameter) by mesh size (60, 200 µm), and stored at −80 °C until ready to process.

Total lipids were extracted with chloroform-methanol (2:1, *v*/*v*) containing 0.01% butylated hydroxytoluene as an antioxidant (*Folch, Lees & Sloane Stanley, 1957*). The organic solvent was evaporated under a stream of nitrogen and lipid concentration determined gravimetrically. Transmethylation of fatty acids was done according to the

method described by *Metcalfe & Schmitz (1961)*. A known amount of nonadecanoate acid (19:0) dissolved in hexane at a concentration of 8 mg ml$^{-1}$ (Nu Check Prep Inc.) was added as internal standard. The fatty acid methyl esters (FAME) were separated by gas chromatography (Agilent 7890A Gas Chromatograph; Agilent Technologies, Inc., Santa Clara, CA, USA) using a 7693 mass spectrometer detector (Agilent Technologies, Inc., Santa Clara, CA, USA), a capillary column (Omegawax$^{TM}$ 250 fused silica capillary column, 30 mm $\times$ 0.25 mm and 0.25 mm film thickness, Supleco$^{\circledR}$), and a 7890A autoinjector (Agilent Technologies, Inc.). Helium was used as the carrier gas at a flow of 1.3 ml min$^{-1}$ and the injection volume was 2 mL. Initial temperature of the oven was 175 °C for 26 min, which was increased to 205 °C by increments of 2 °C min$^{-1}$, then held at 205 °C for 24 min. The source and analyzer for the mass spectrometer was set at 230 °C. The individual fatty acid methyl esters were identified by comparing the retention times of authentic standard mixtures (FAME mix 37 components, Supleco) and quantified by comparing their peak areas with that of the internal standard (*Czesny & Dabrowski, 1998*). The results of individual fatty acid composition are expressed in percentage of total identified FAME.

*Statistical analysis.* We performed a series of multivariate analyses to address our specific objectives. We used PERMANOVA a part of the PRIMER 6 statistical software package (*Clarke & Gorley, 2006*), to test for overall differences between the microzooplankton and mesozooplankton community composition, and fatty acid profiles of seston, microzooplankton, and mesozooplankton. PERMANOVA is a non-parametric technique related to ANOVA, but uses permutations and fewer assumptions compared to the traditional ANOVA approach (*Anderson, 2001*). As such, it is particularly well suited to multivariate data sets that violate the traditional assumptions of ANOVA and also have low sample sizes, as was our case (*Anderson, 2001*).

If PERMANOVA detected differences, we then generated separate, Bray-Curtis similarity matrices for microzooplankton species composition (60 μm mesh), mesozooplankton species composition (200 μm mesh), seston fatty acid composition, microzooplankton fatty acid composition, and mesozooplankton fatty acid composition. A separate cluster analysis was performed using PRIMER 6 in order to reveal patterns over time and space for each similarity matrix. Each individual sample was associated with a location in the river (upper, middle, lower) and month (April, May, June) and these labels were used for visualization of samples in the cluster dendrogram. Next, a similarity percentage analysis (SIMPER) test was used to compare similarities within groups and determine the species or fatty acids that contributed to each grouping from the cluster analysis (*Clarke & Gorley, 2006*). The SIMPER test was set at 70% cumulative contribution.

We then wanted to determine if salinity and temperature were related to the observed patterns and we used redundancy analysis for this purpose (*Legendre & Legendre, 1998*). The redundancy analysis was carried out in the R environment (R v3.2.3, *R Core Development Team, 2015*) using rda function in the vegan package (*Oksanen et al., 2017*). Finally, we used a Mantel matrix comparison to correlate fatty acid profiles between the three groups (seston, microzooplankton, and mesozooplankton). The mantel.rtest function in the ade4

**Table 1  Results of SIMPER analysis for each group; all the zooplankton species that contributed community composition, and their corresponding values in % are given.** Dash marks represent species that were not included in the contribution of 70%.

|  | Microzooplankton | | Mesozooplankton | | |
|---|---|---|---|---|---|
|  | Group 1 | Group 2 | Group 3 | Group 4 | Group 5 |
| Overall similarity | 91.18 | 80.18 | 84.39 | 57.16 | 52.52 |
| Species | Percent composition | | Percent composition | | |
| Bosminidae | – | – | 17.26 | 34.25 | 12.20 |
| *Leptodora* spp. | – | – | – | – | 21.80 |
| Calanoida | – | – | – | 33.32 | 26.09 |
| Cyclopoida | – | – | – | 10.24 | 9.07 |
| *Acartia* spp. | – | – | 74.66 | – | – |
| Copepod nauplii | 10.60 | 41.61 | – | – | – |
| Rotifer | 86.47 | 50.46 | – | – | – |

package (*Oksanen et al., 2017*) in the R environment (R v3.2.3, *R Core Development Team, 2015*) was used.

# RESULTS

## Salinity and temperature

Salinity was near zero (0.02–0.04) throughout the river in April. During May, salinities in the upper river remained low (0.07), but the water column became stratified in the middle and lower river, with salinities ranging from 0.28–1.66. The river returned to freshwater (0.04–0.08) again in June due to a tropical storm that brought heavy rains for a two-week period. North Carolina experienced the second wettest June since 1895 with rainfall that ranged from 15.2 to 19.05 cm in the study area (*Hiatt, 2013*). Water temperatures increased during the study period April ($15.7 \pm 1.1\,°C$), May ($24.0 \pm 1.0\,°C$) and June ($26.2 \pm 0.3\,°C$).

### Zooplankton community composition

There were significant differences between the overall microzooplankton and mesozooplankton community composition (PERMANOVA, $p = 0.001$). Microzooplankton could be separated into two distinct groups by cluster analysis at 65% similarity (Fig. 2A). Group 1 consisted of the vast majority of the samples collected across April, May, and June throughout the river (Fig. 2A). Group 2 consisted of two samples collected in the middle and lower river in June (Fig. 2A). Both groups were dominated by rotifers and copepod nauplii, but group 2 had a higher contribution of copepod nauplii (Fig. 2B and Table 1).

Three groups of mesozooplankton were differentiated at 50% similarity using cluster analysis (Fig. 3A). Group 3 consisted of samples from the May collection only, Group 4 consisted of a mixture of April and June samples in primarily the upper and middle river, and Group 5 consisted of one April upper site, May upper river section and June samples throughout the river (Fig. 3A). Group 3 mesozooplankton percent composition was dominated by *Acartia* spp. with a minor contribution by Bosminidae, Group 4 consisted primarily of equivalent percentages of Cyclopoida and Bosminidae, and

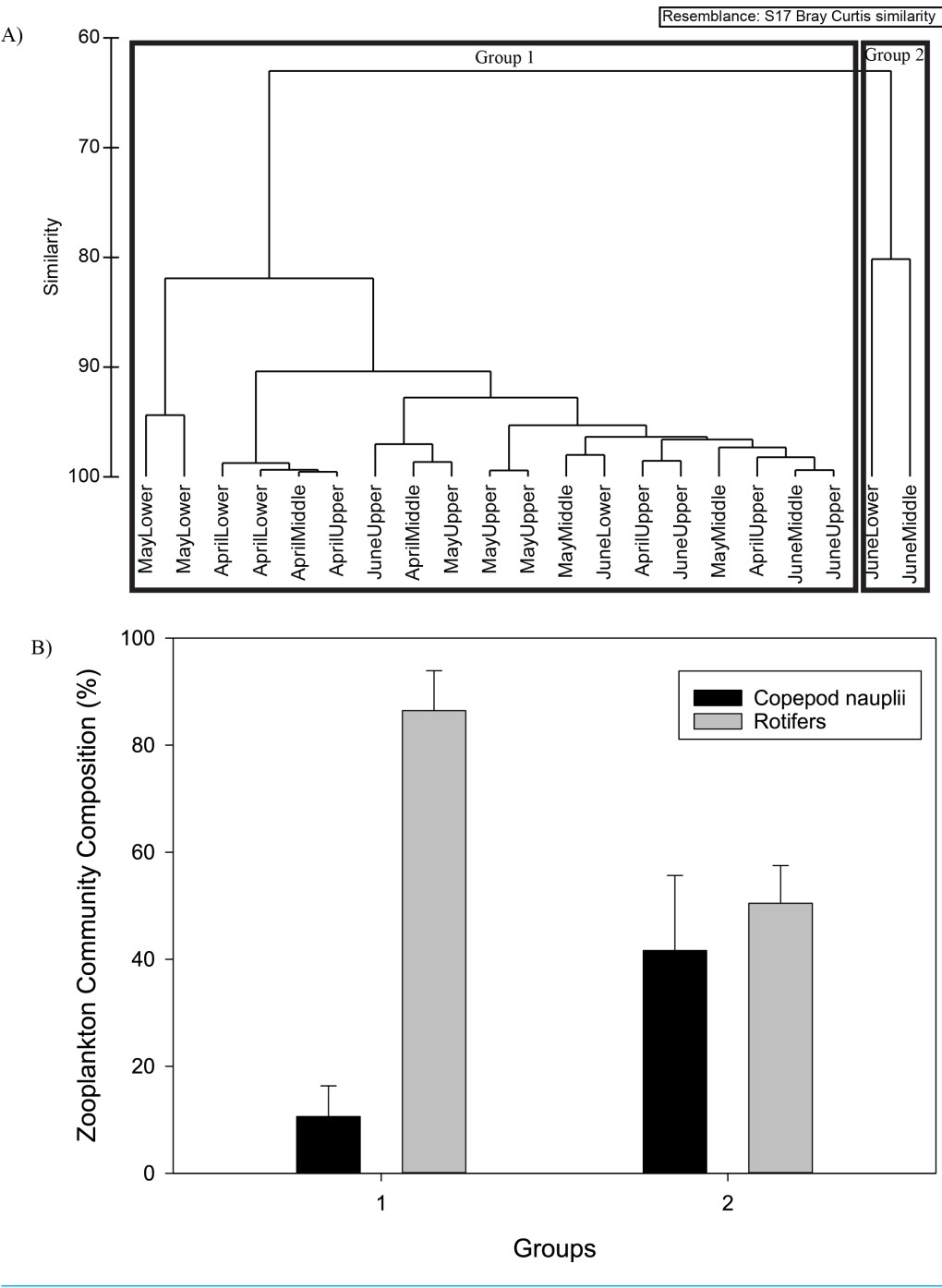

**Figure 2** The two microzooplankton community composition groups from cluster analysis (A) at 65% similarity. The mean microzooplankton community composition (%, ±S.D.) for the two groups from cluster analysis (B).

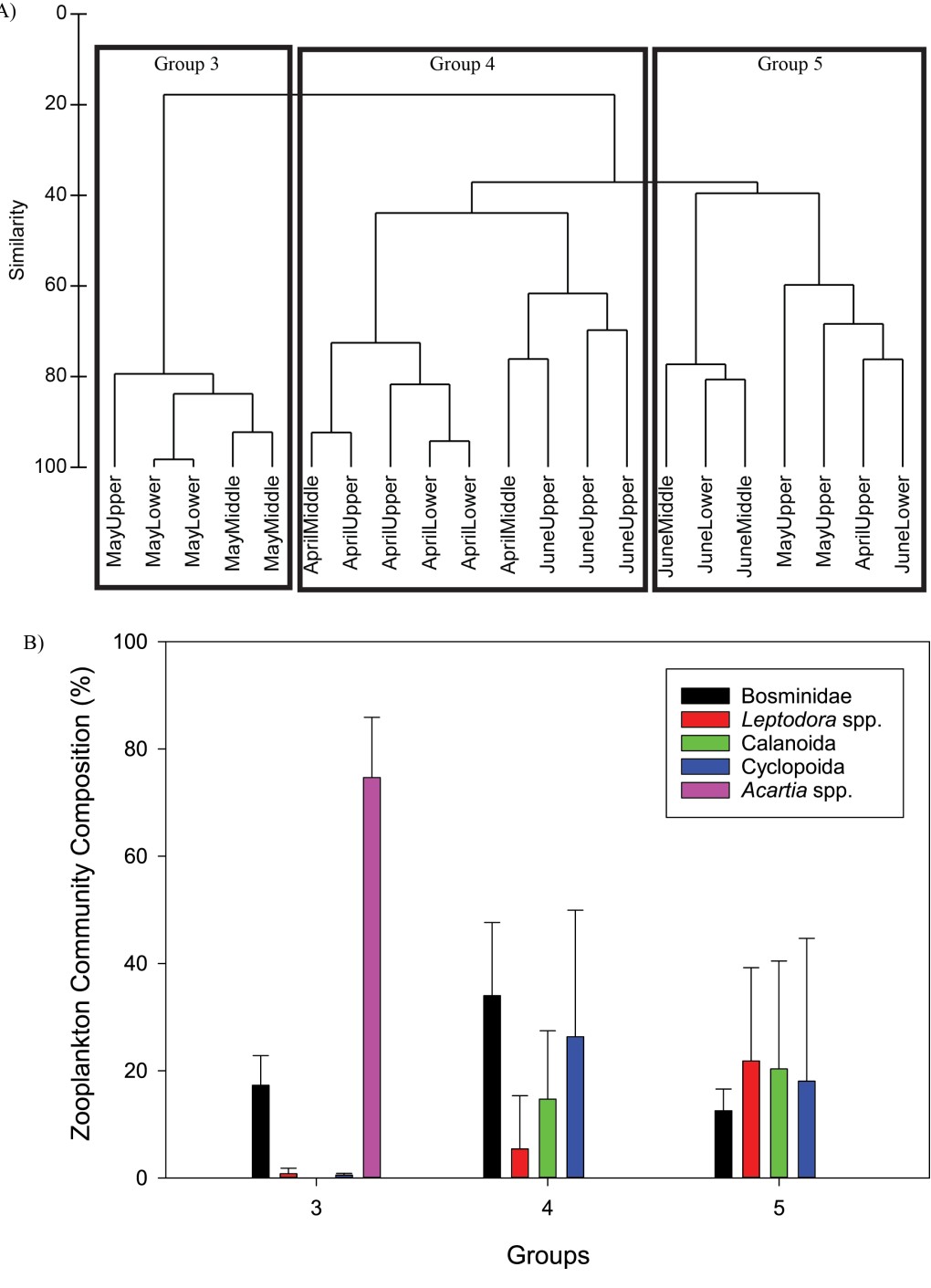

**Figure 3** The three mesozooplankton community composition groups from cluster analysis (A) at 50% similarity. The mean mesozooplankton community composition (%, ±S.D.) for the three groups from cluster analysis (B).

Table 2 **Results of SIMPER analysis for each group; the fatty acids that contributed to the differences in fatty acid profile, and their corresponding values in % are given.** Dash marks represent fatty acids that were not included in the contribution of 70%. Group G had a sample size <2.

| | Seston | | | Microzooplankton | | | | Mesozooplankton | | |
|---|---|---|---|---|---|---|---|---|---|---|
| | Group A | Group B | Group C | Group D | Group E | Group F | Group G | Group H | Group I | Group J |
| Overall similarity | 66.40 | 72.32 | 64.34 | 84.88 | 92.49 | 81.05 | N/A | 81.42 | 89.42 | 84.89 |
| Fatty acids | Percent composition | | | Percent composition | | | | Percent composition | | |
| 16:1ω-7 | 3.12 | 6.53 | – | – | – | 7.98 | – | 9.53 | – | 8.51 |
| 18:1ω-9 | 7.04 | 2.27 | 2.42 | 31.63 | 13.46 | – | 17.01 | 10.58 | – | – |
| 18:2ω-6 | – | – | – | 12.15 | – | – | – | – | – | – |
| 18:3ω-3 (ALA) | 2.37 | – | 3.33 | – | 8.43 | 6.80 | – | 6.14 | 10.65 | – |
| 18:4ω-3 | – | – | – | – | 10.29 | – | – | – | 7.05 | – |
| 20:5ω-3 (EPA) | – | 1.55 | 2.44 | – | 11.72 | 9.91 | – | 11.24 | 13.84 | 16.40 |
| 22:6ω-3 (DHA) | – | – | 1.87 | – | – | 9.45 | – | – | 10.11 | 16.58 |

Group 5 mesozooplankton percent community composition was characterized by higher percentages of *Leptodora* spp. and Calanoida (Fig. 3B and Table 1).

### Fatty acid composition

A total of 24 specific fatty acids were found in all samples (Tables A1–A3). Fatty acids were first separated into broad categories: saturated fatty acids (SFA), monounsaturated fatty acids (MUFA), and polyunsaturated fatty acids (PUFA) (Fig. 4A). Seston had a higher percent of SFA, and lower percent of MUFA and PUFA compared to micro- and mesozooplankton (Fig. 4A). Mesozooplankton and microzooplankton had a similar percent composition of SFA, MUFA, and PUFA (Fig. 4A). There were eight dominant fatty acids found in all the samples, but the percent composition varied (Fig. 4B). The most common SFA was palmitic acid (16:0), the most common MUFAs were palmitoleic acid (16:1ω-7) and oleic acid (18:1ω-9), and the most common PUFAs were ALA, 18:4ω-3, EPA, and DHA (Fig. 4B; Tables A1–A5). A comparison of MUFAs and PUFAs among the seston and zooplankton showed that seston had the lowest overall percentages of MUFAs and PUFAs (PERMANOVA, $p = 0.001$, Fig. 4B). There was a difference between the microzooplankton fatty acid profile and the mesozooplankton fatty acid profile (PERMANOVA, $p = 0.025$). The microzooplankton fatty acid profile was characterized by a higher percentage of 18:1ω-9 compared to the other MUFAs and PUFAs. In contrast, the mesozooplankton had the highest percent composition attributed to two PUFAs, EPA and DHA (Fig. 4B).

### Seston

Three groups were designated at 60% similarity using cluster analysis for the seston fatty acid composition (Fig. 5A). The groups showed no distinct pattern in terms of sampling time or location (Fig. 5A). Seston fatty acid composition of Group A was characterized by 18:1ω-9, 16:1ω-7, ALA, and EPA, Group B by 16:1ω-7, 18:1ω-9, 18:2ω-6 and EPA, and Group C had similar percentage composition of MUFAs and PUFAs, except 18:2ω-6 (Fig. 5B and Table 2).

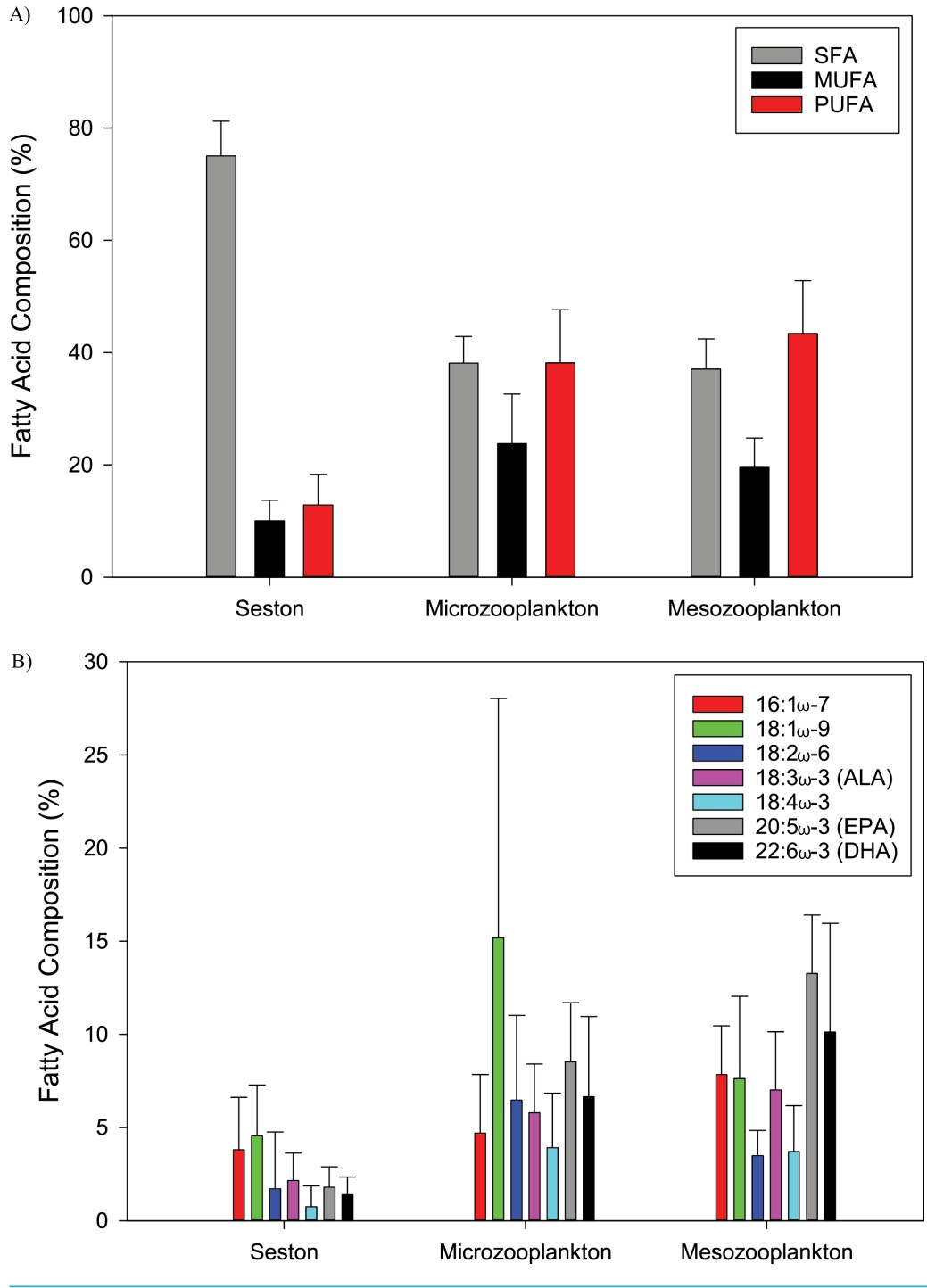

**Figure 4** The mean (±S.D.) saturated (SFA), monounsaturated (MUFA), and polyunsaturated (PUFA) fatty acid composition (%) for the seston, microzooplankton, and mesozooplankton (A). The mean fatty acid composition (%, ±S.D.) for the seston, microzooplankton, and mesozooplankton (B).

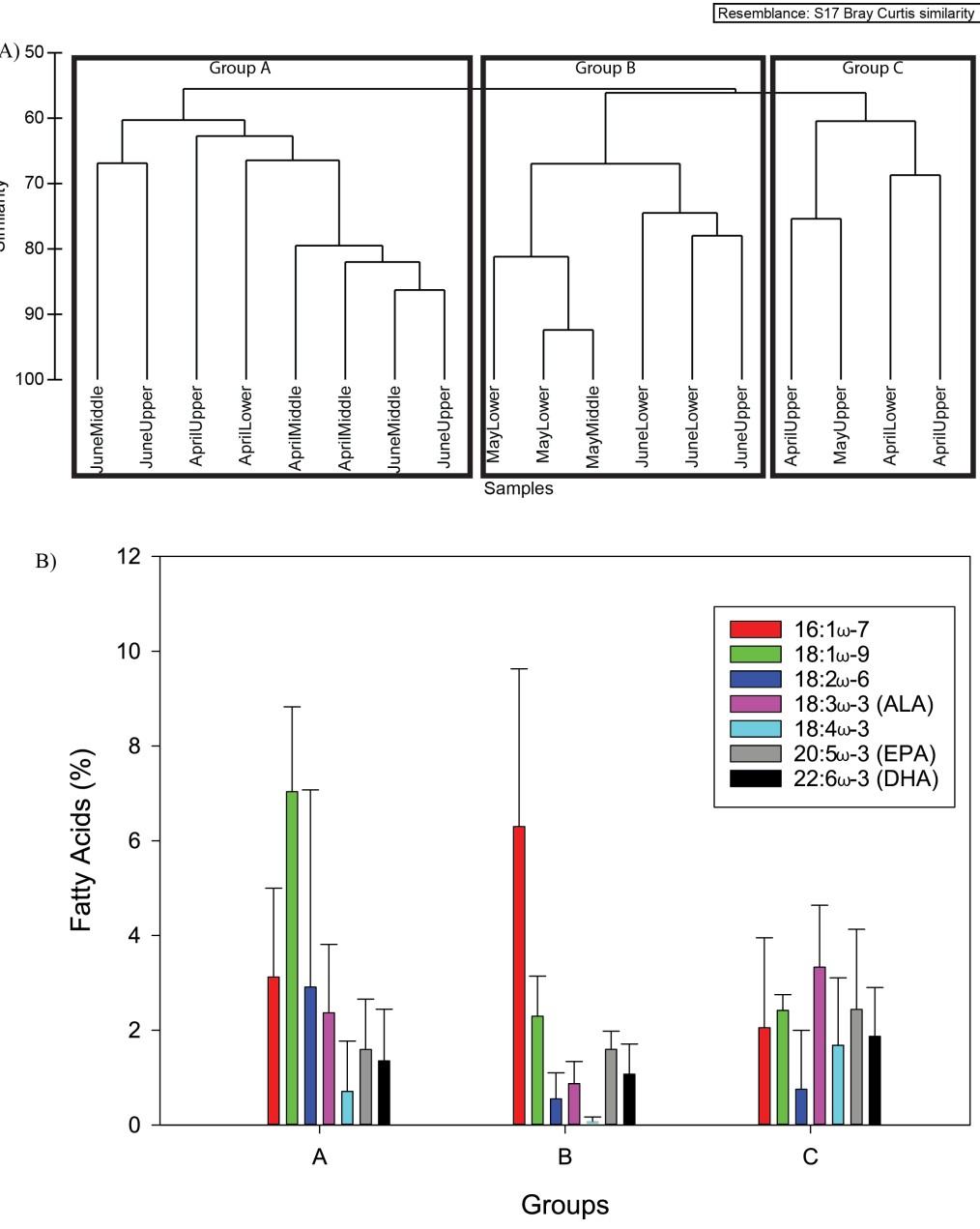

**Figure 5** The three seston fatty acid composition groups from cluster analysis (A) at 60% similarity. The mean seston fatty acid composition (%, ±S.D.) for the three groups from cluster analysis (B).

## Microzooplankton

Three groups were designated at 70% similarity using cluster analysis for the microzooplankton fatty acid composition (Fig. 6A). The groups segregated temporally, with Group D and E consisting of April samples only, and Group F consisted of May and June samples only (Fig. 6A). The Group D fatty acids were dominated by 18:1ω-9 and to a lesser extent, 18:2ω-6, Group E showed similar percent composition among the fatty acids, with higher percentages of 18:1 ω-9 and EPA, and Group F also had similar percent

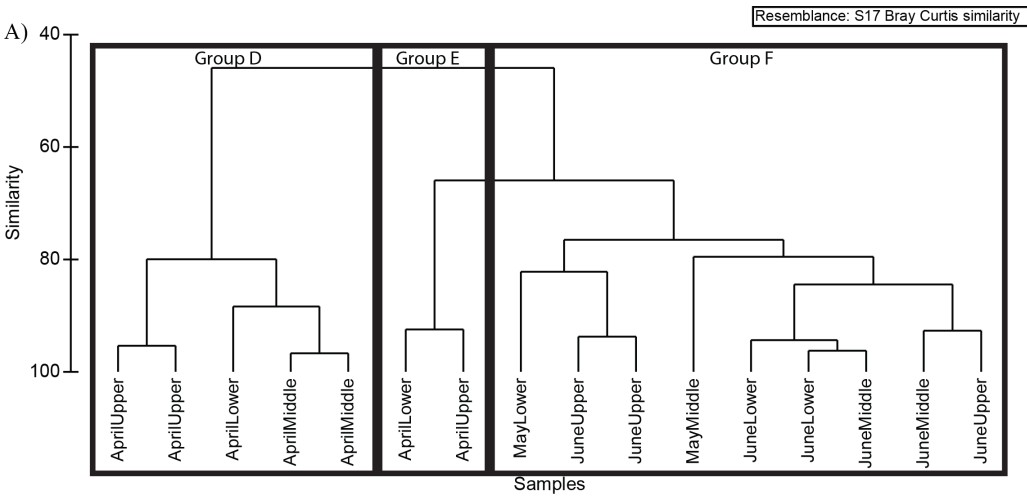

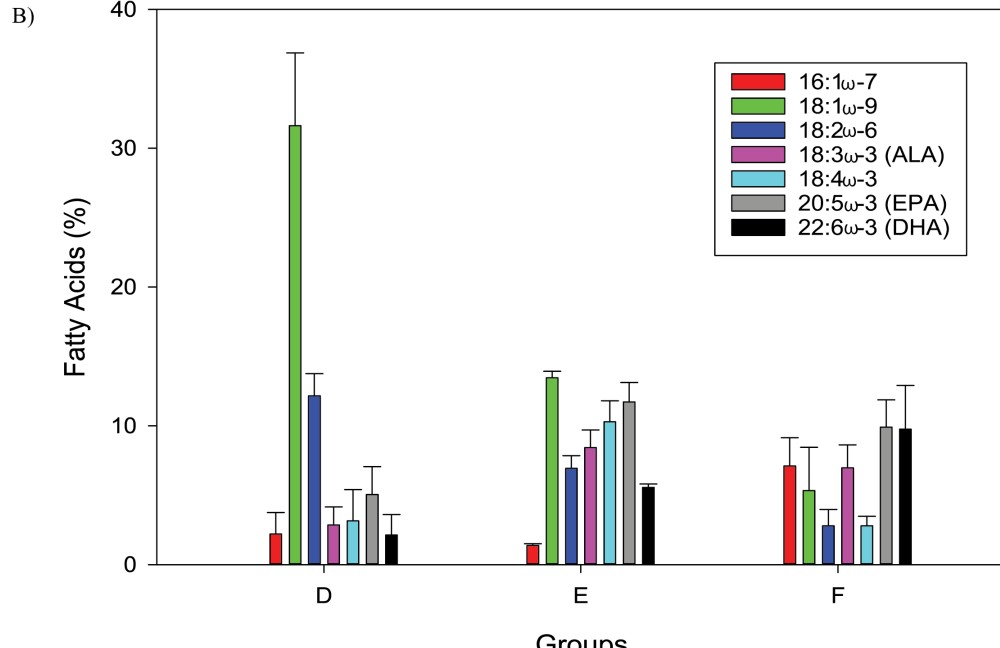

**Figure 6** The three microzooplankton fatty acid composition groups from cluster analysis (A) at 60% similarity. The mean microzooplankton fatty acid composition (%, ±S.D.) for the three groups from cluster analysis (B).

composition of fatty acids; however, the PUFAs EPA and DHA had significant contribution (Fig. 6B and Table 2).

### Mesozooplankton

Four groups were designated by cluster analysis at 77% similarity for the mesozooplankton fatty acid composition (Fig. 7A). Groups G and I consisted of April samples only whereas Group J consisted of May and June samples only (Fig. 7A). Group H showed spatial

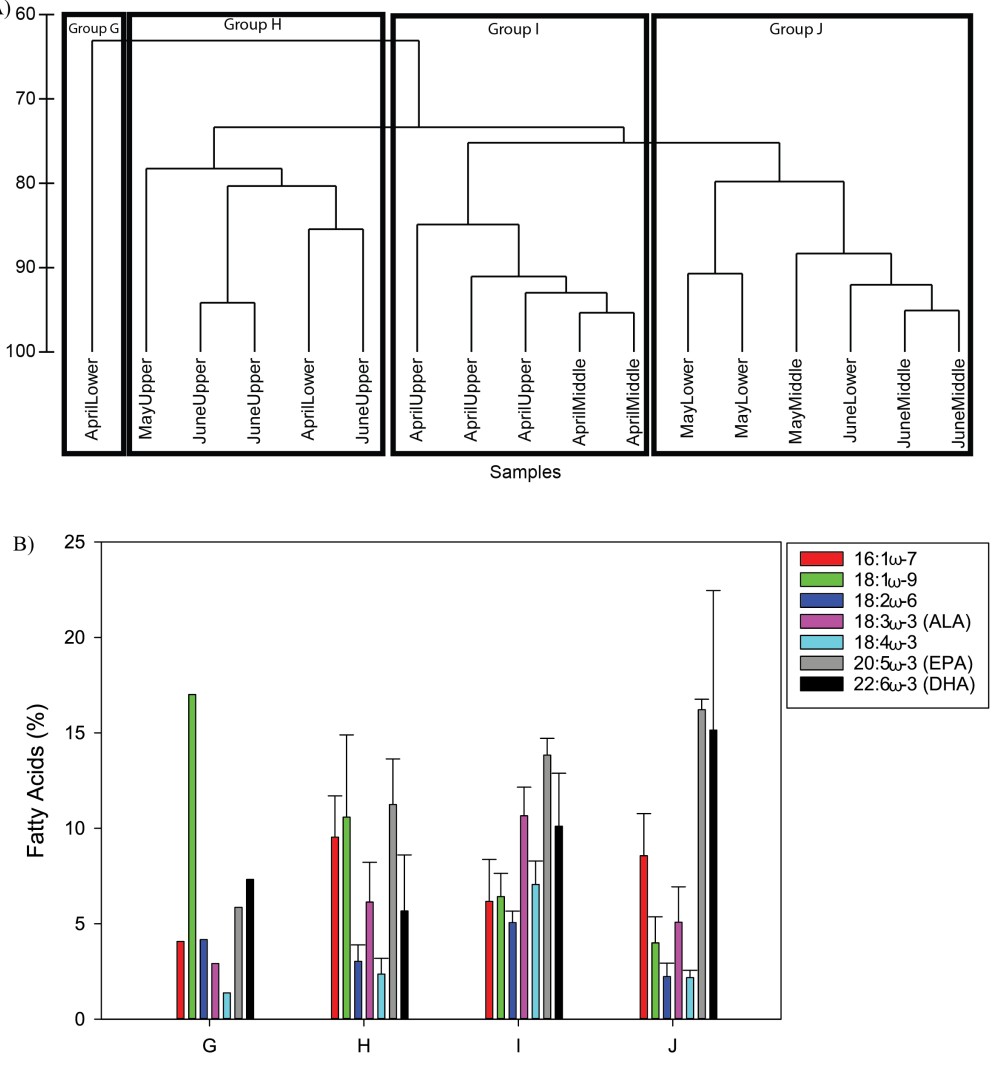

**Figure 7** The four mesozooplankton fatty acid composition groups from cluster analysis (A) at 60% similarity. The mean mesozooplankton fatty acid composition (%, ±S.D.) for the four groups from cluster analysis (B).

separation, consisting of primarily upper river locations across all of the months (Fig. 7A). The Group G and H fatty acids were dominated by 18:1ω-9, Group 8 by 16:1 ω-7, 18:1ω-9, and EPA, Group I had similar percentages of fatty acids with ALA, DHA and EPA having higher percentages, and Group J had DHA and EPA as dominant components of the fatty acids (Fig. 7B and Table 2).

### Redundancy analysis and Mantel matrix correlations

Salinity was the main factor found to be correlated to observed patterns in the species composition of both microzooplankton and mesozooplankton (Redundancy, $p = 0.004$).

However, salinity was not correlated to changes in the seston (Redundancy, $p = 0.490$) or microzooplankton (Redundancy, $p = 0.27$) fatty acid profiles. Salinity was associated with the changes in the mesozooplankton fatty acid profiles (Redundancy, $p = 0.034$). Seston fatty acid profiles were correlated to the microzooplankton fatty acid profiles ($p = 0.013$) based on the Mantel matrix comparison, but not correlated to the mesozooplankton fatty acid profiles ($p = 0.340$). The microzooplankton fatty acid profiles were correlated to the mesozooplankton fatty acid profiles (Mantel, $p = 0.059$).

## DISCUSSION

We found temporal and spatial differences in the species and fatty acid composition of the lower food web that were mainly related to a salinity intrusion that occurred during the study period during May. Prior to the salinity intrusion, larval fish would have encountered a freshwater plankton assemblage that was dominated by rotifers, Bosminidae, and cylopoid copepods throughout the river. This assemblage was proportionally higher in EPA relative to DHA. During the salinity intrusion, the microzooplankton remained dominated by rotifers; however, the mesozooplankton community became dominated by the copepod *Acartia* spp. Concurrently, the proportion of DHA increased and remained elevated into June, particularly in the middle and lower river. The intrusion of saline water increased the overall proportion of omega-3 fatty acids in the river, presumably due to the increased fraction of micro- and mesozooplankton feeding on a more marine-like phytoplankton based food web and this signal propagated through the food web. Additionally, we observed that FA appeared to be incorporated relatively unchanged in micro and mesozooplankton in terms of relative composition; however, MUFA and PUFA percent compositions increased in zooplankton relative to seston. This suggested that MUFA and PUFA are bioaccumulated at higher trophic levels, as seen in other studies (*Persson & Vrede, 2006*; *Gladyshev et al., 2010*; *Ravet, Brett & Arhonditsis, 2010*; *Burns, Brett & Schallenberg, 2011*). Overall, the FA composition of the food web indicated that the Chowan River is likely to provide nutrition in terms of FA composition for larval fish growth and development. This is based on the presence of higher chain (>20 carbons) PUFAs present in the mesozooplankton throughout the nursery.

The seston fatty acid composition consisted mainly of saturated fatty acids. Seston from freshwater and estuarine systems typically has a large percentage of SFA and this fraction has been attributed to detrital input, as opposed to originating from phytoplankton (*Persson & Vrede, 2006*; *Gladyshev et al., 2010*; *Ravet, Brett & Arhonditsis, 2010*; *Burns, Brett & Schallenberg, 2011*; *Goncalves et al., 2012*). *Müller-Navarra et al. (2004)* and *Bec et al. (2010)* analyzed seston and found phytoplankton only explained 27% of variance in FA composition and concluded that detritus and heterotrophic organisms also needed to be considered. *Bec et al. (2010)* therefore concluded that the seston can affect the fatty acid profiles of higher organisms, but may not relate individual groups of phytoplankton or microzooplankton. This agreed with our findings as seen in the reduced correlations between seston and the mesozooplankton.

We did not examine the seston composition directly by counting phytoplankton cells or examining pigment concentrations, thus we were unable to attribute the origin of

particular fatty acids to phytoplankton or other sources. However, we were able to use the available literature to identify potential indicators of fatty acid origin. The top four fatty acids by percent composition varied by group, but 16:1ω-7, 18:1ω-9, 18:2 ω-6, and ALA were the most prevalent. Potential phytoplankton sources for these fatty acids may be diatoms, which have been shown to have increased 16:1ω-7 and EPA in both freshwater and marine systems (*Napolitano et al., 1997*; *Dalsgaard et al., 2003*; *Boschker, Krombamp & Middelburg, 2005*; *Arts, Brett & Kainz, 2009*; *Bec et al., 2010*) and we observed this occurred in May and June coincident with the salinity increase. Green algae (Chlorophytes) have been shown to possess higher proportions of 18:2ω-6, and ALA, and 18:4ω-3 (*Ahlgren et al., 1990*; *Dalsgaard et al., 2003*; *Boschker, Krombamp & Middelburg, 2005*; *Masclaux et al., 2012*; *Strandberg et al., 2015*). Fatty acids corresponding to these phytoplankton groups were observed during April throughout the river and June in the middle and upper river. One other source of seston FA may have been present, pine pollen, which is found in large quantities during spring. Pine pollen is transported to freshwater systems via aerial deposition and floats at the surface, and the pine pollen fatty acid profile has a high percent composition of 18:1ω-9 and 18:2ω-6 (*Masclaux et al., 2013*), which can be observed in our samples. Obviously, seston FA are a mixture of multiple sources, thus the variability seen across the groups identified by the cluster analysis would be expected.

The microzooplankton fatty acid profiles were different throughout the sampling period with a change from decreased omega-3s to increased omega-3s in the system. This suggests a switch in microzooplankton diet had occurred over the sampling period and two pathways appear to be present during the study. The April microzooplankton fatty acid profiles for all river sections had a high percentage of 18:1ω-9 and 18:2ω-6 suggesting that the microzooplankton could be consuming either terrestrial material or chlorophytes during this time. Two sites in April had an increase in omega-3 fatty acids (ALA, 18:4 ω-3, EPA, DHA), and this would suggest a different dietary pathway that was reduced in SFA, perhaps consisting of either smaller microzooplankton such as ciliates or phytoplankton such as diatoms and/or dinoflagellates (*Park & Marshall, 2000*; *Gladyshev et al., 2010*). The community was dominated by rotifers during this time and communities high in rotifer abundance have been shown to closely reflect the seston composition (*Gladyshev et al., 2010*). The microzooplankton fatty acid profiles in May and June at all river locations had an increase in 16:1ω-7, and omega-3s (ALA, EPA, and DHA). These changes can be correlated to the likely increase in diatoms and dinoflagellates from the saltwater intrusion event. The changes in June could be the increase in copepod nauplii of Calanoid copepods, and the presence of dinoflagellates and diatoms even when the system returned to freshwater. Our results are similar to systems where the phytoplankton composition was represented by diatoms and dinoflagellates by having increased 16:1ω-7 and PUFAs (*Müller-Navarra et al., 2000*; *Dalsgaard et al., 2003*; *Gladyshev et al., 2010*; *Ravet, Brett & Arhonditsis, 2010*).

The mesozooplankton fatty acid profiles throughout the river in April and in the upper river in May and June were defined by higher percentages of 16:1ω-7, 18:1ω-9, ALA, EPA, and DHA, and mixed mesozooplankton community consisting of cladoceran and copepods. These fatty acids profiles are similar to those found in freshwater systems that have a mixed zooplankton composition (*Persson & Vrede, 2006*; *Arts, Brett & Kainz, 2009*;

*Kainz et al., 2009*; *Gladyshev et al., 2010*; *Burns, Brett & Schallenberg, 2011*; *Masclaux et al., 2012*). Cladocerans have low or no DHA compared to copepods and high EPA levels have been shown to correlate with the high somatic growth rates of cladoceran (*Persson & Vrede, 2006*). A saltwater intrusion changed the mesozooplankton species composition in May resulting in numerical dominance by *Acartia* spp. in the lower and middle sections of the river. *Acartia* spp. is the dominant copepod species in temperate, estuarine systems (*Ambler, Cloern & Hutchinson, 1985*; *Orsi & Mecum, 1986*; *Cervetto, Gaudy & Pagano, 1999*; *Mouny & Dauvin, 2002*; *Kimmel & Roman, 2004*; *Lawrence, Valiela & Tomasky, 2004*; *Islam, Ueda & Tanaka, 2005*). The mesozooplankton fatty acid profiles in May were represented by 16:1ω-7, EPA, and the highest observed percentages of DHA. This is clearly a reflection of the dominance of *Acartia* spp. in the system and a diet primarily consisting of marine algae higher in omega-3 FAs (*Stottrup, Bell & Sargent, 1999*; *Persson & Vrede, 2006*; *Arts, Brett & Kainz, 2009*; *Kainz et al., 2009*; *Gladyshev et al., 2010*; *Masclaux et al., 2012*). Mesozooplankton fatty acid percent composition in June at the lower and middle site remained similar to that observed in May, despite the species composition having returned to a mix of cladocerans and copepods. This suggests that physical shifts in the system that result in seston changes may persist in the system despite shifts in zooplankton community composition.

The relevance of the food web fatty acid composition can be determined by examining the potential feeding behavior of larval fish within the Chowan River nursery. Alewife and blueback herring start feeding on smaller cladocerans and copepods at about 6 mm total length (*Mullen, Fay & Moring, 1986*). *Binion (2011)* reported that river herring at 6 mm notochord length had a maximum gape width of 400 $\mu$m, and estimated maximum prey size of 200 $\mu$m, which would result in mesozooplankton being an important food resource. In the Connecticut River, the diet for blueback herring were dominated by rotifers for fish 5–12 mm, Bosminidae for fish 12–16 mm, and cyclopoid copepods for fish >16 mm in total length (*Crecco & Blake, 1983*). Based on these dietary studies, the larval and juvenile river herring would be feeding across the size range of zooplankton prey that we sampled; however, fish would be consuming primarily microzooplankton early in the year (April) and mesozooplankton later in the season (May and June). In April, two pathways for FA propagation were present in the microzooplankton. Thus, fish feeding during this time may experience variability in the quality of the microzooplankton prey in terms of percentage of PUFAs. Larval fish need PUFAs (ALA, EPA and DHA) for growth, visual acuity, survival, and development of normal pigmentation (*Bell et al., 1995*; *Bell & Sargent, 1996*; *Rainuzzo, Reitan & Olsen, 1997*; *Sargent et al., 1999*; *Rossi et al., 2006*). The shift to larger prey later in the year (May and June) resulted in a change in prey quality as the relative percentage of EPA and DHA increased. This was the result of a salinity intrusion into the middle and lower reaches of the estuary that was associated with dominance of the cladoceran *Acartia* spp. and a significant increase in DHA and EPA. This could allow larval and juvenile fish to consume prey with a higher proportion of PUFAs. It is unknown if river herring can elongate precursor FA into PUFAs. Even if fish can convert precursor FA, larval fish could not receive all nutritional need for those fatty acids (*Agaba et al., 2005*). The larval fish would not have to use energy for the conversion, and could continue to put energy into

growth (*Wacker & Von Elert, 2001*; *Rossi et al., 2006*). This allows the larval fish to survive and grow past the critical period.

The fish nursery present in the lower Chowan River may undergo significant changes during the critical time of larval fish growth and our results demonstrate how changes in the seston community may propagate through the food web. The results also highlight that additional information concerning the fatty acid composition of the zooplankton prey base for larval fish can provide insight into habitat quality, our stated goal. *Sheaves et al. (2015)* pointed out the need to expand the nursery habitat concept to include relevant ecosystem processes, particularly resource dynamics. This research begins to explore the mechanisms that allow nursery habitat to function. We plan further research to investigate the linkage between the seston community fatty acid composition, the zooplankton community fatty acid composition, and larval fish to determine how lower food web variability relates to larval fish condition and survival.

## ACKNOWLEDGEMENTS

I would like to thank S Lichti, C Krahforst, J Osborne, A Powell, M Baker, and E Diaddorio for help in the field collection. I would like to thank L Stratton, C Kolb, and R Pattridge for help in Dr. Jacques Rinchard's lab in processing my fatty acid samples. I would like to thank Dr. Ariane Peralta for help with the improved statistical analysis, and two anonymous reviewers for comments, which helped to improve the manuscript.

## APPENDIX

**Table A.1** **Mean fatty acid composition (±standard deviation) (percentage of total fatty acids detected) of seston from the Chowan River by group.**

| | Seston group | | |
|---|---|---|---|
| | **A (8)** | **B (5)** | **C (4)** |
| 14:0 | $6.6 \pm 1.8$ | $11.2 \pm 5.1$ | $7.6 \pm 1.0$ |
| 15:0 | $1.5 \pm 1.5$ | $1.3 \pm 0.4$ | $1.7 \pm 0.3$ |
| 16:0 | $47.7 \pm 5.5$ | $54.0 \pm 5.1$ | $50.8 \pm 4.4$ |
| 17:0 | $2.3 \pm 0.4$ | $2.7 \pm 0.7$ | $2.6 \pm 0.2$ |
| 18:0 | $14.2 \pm 5.6$ | $9.7 \pm 1.2$ | $11.4 \pm 2.5$ |
| 20:0 | $0.4 \pm 0.3$ | $0.3 \pm 0.1$ | $0.4 \pm 0.2$ |
| $\sum$**SFA** | **72.7** | **79.2** | **74.0** |
| 16:1$\omega$-9 | $1.2 \pm 1.0$ | $1.3 \pm 0.6$ | $1.6 \pm 0.7$ |
| 16:1$\omega$-7 | $3.1 \pm 1.9$ | $6.3 \pm 3.3$ | $2.1 \pm 1.9$ |
| 18:1$\omega$-9 | $7.0 \pm 1.8$ | $2.3 \pm 0.8$ | $2.4 \pm 0.3$ |
| 18:1$\omega$-7 | $0.3 \pm 0.2$ | $0.1 \pm 0.2$ | $0.1 \pm 0.1$ |
| 20:1 | $0.1 \pm 0.0$ | $0.1 \pm 0.0$ | $0.2 \pm 0.1$ |
| $\sum$**MUFA** | **11.7** | **10.1** | **6.4** |

**Table A.1** (*continued*)

| | Seston group | | |
|---|---|---|---|
| | A (8) | B (5) | C (4) |
| 18:2ω-6 | 2.9 ± 4.2 | 0.6 ± 0.6 | 0.8 ± 1.2 |
| 18:3ω-3 | 2.4 ± 1.4 | 0.9 ± 0.5 | 3.3 ± 1.3 |
| 18:4ω-3 | 0.7 ± 1.1 | 0.1 ± 0.1 | 1.7 ± 1.4 |
| 20:2ω-6 | 0.5 ± 0.4 | 0.2 ± 0.2 | 0.7 ± 0.5 |
| 20:3ω-6 | 0.4 ± 0.1 | 0.4 ± 0.3 | 0.3 ± 0.2 |
| 20:4ω-6 | 0.9 ± 0.6 | 0.8 ± 0.4 | 0.8 ± 0.5 |
| 20:3ω-3 | 0.6 ± 0.3 | 0.4 ± 0.3 | 1.1 ± 0.6 |
| 20:4ω-3 | 0.6 ± 1.4 | 0.5 ± 0.3 | 2.3 ± 1.9 |
| 20:5ω-3 | 1.6 ± 1.1 | 1.6 ± 0.4 | 2.4 ± 1.7 |
| 22:5ω-6 | 0.7 ± 0.5 | 0.8 ± 0.8 | 1.2 ± 0.6 |
| 22:5ω-3 | 0.8 ± 0.6 | 0.7 ± 0.6 | 1.2 ± 0.7 |
| 22:6ω-3 | 1.4 ± 1.1 | 1.1 ± 0.6 | 1.9 ± 1.0 |
| $\sum$PUFA | 13.5 | 8.1 | 17.7 |

Notes.

SFA, saturated fatty acids; MUFA, monounsaturated fatty acids; PUFA, polyunsaturated fatty acids..

**Table A.2  Mean fatty acid composition (percentage of total fatty acids detected) of microzooplankton (<60 μm) from the Chowan River by groups.**

| | Microzooplankton Groups | | |
|---|---|---|---|
| | D (5) | E (2) | F (8) |
| 14:0 | 2.6 ± 0.5 | 3.7 ± 0.9 | 5.9 ± 1.1 |
| 15:0 | 0.3 ± 0.1 | 0.5 ± 0.2 | 1.1 ± 0.3 |
| 16:0 | 27.4 ± 3.0 | 24.5 ± 0.3 | 23.5 ± 3.4 |
| 17:0 | 0.4 ± 0.1 | 0.6 ± 0.0 | 1.2 ± 0.3 |
| 18:0 | 3.4 ± 0.8 | 3.4 ± 0.0 | 7.1 ± 0.3 |
| 20:0 | 1.8 ± 0.4 | 0.8 ± 0.1 | 0.2 ± 0.1 |
| $\sum$SFA | 35.9 | 33.5 | 39.0 |
| 16:1ω-9 | 0.3 ± 0.1 | 0.8 ± 0.3 | 0.9 ± 0.6 |
| 16:1ω-7 | 2.2 ± 1.6 | 1.4 ± 0.1 | 7.1 ± 2.0 |
| 18:1ω-9 | 31.6 ± 5.3 | 13.5 ± 0.5 | 5.3 ± 3.1 |
| 18:1ω-7 | 0.7 ± 0.8 | 0.8 ± 0.2 | 1.9 ± 0.5 |
| 20:1 | 0.6 ± 0.2 | 0.8 ± 0.1 | 1.4 ± 0.6 |
| $\sum$MUFA | 35.4 | 17.3 | 16.6 |
| 18:2ω-6 | 12.2 ± 1.6 | 6.9 ± 0.9 | 2.8 ± 1.2 |
| 18:3ω-3 | 2.9 ± 1.3 | 8.4 ± 1.3 | 7.0 ± 1.7 |
| 18:4ω-3 | 3.2 ± 2.3 | 10.3 ± 1.5 | 2.8 ± 0.7 |
| 20:2ω-6 | 0.4 ± 0.1 | 0.5 ± 0.1 | 0.3 ± 0.1 |
| 20:3ω-6 | 0.1 ± 0.1 | 0.1 ± 0.0 | 0.2 ± 0.1 |
| 20:4ω-6 | 0.6 ± 0.7 | 0.4 ± 0.1 | 3.2 ± 0.5 |

**Table A.2** (*continued*)

| | Microzooplankton Groups | | |
|---|---|---|---|
| | **D (5)** | **E (2)** | **F (8)** |
| 20:3$\omega$-3 | 0.3 $\pm$ 0.1 | 1.0 $\pm$ 0.2 | 0.4 $\pm$ 0.1 |
| 20:4$\omega$-3 | 1.3 $\pm$ 0.4 | 2.8 $\pm$ 0.1 | 2.1 $\pm$ 0.3 |
| 20:5$\omega$-3 | 5.0 $\pm$ 2.0 | 11.7 $\pm$ 1.4 | 9.9 $\pm$ 2.0 |
| 22:5$\omega$-6 | 0.2 $\pm$ 0.2 | 1.0 $\pm$ 0.1 | 2.3 $\pm$ 0.9 |
| 22:5$\omega$-3 | 0.2 $\pm$ 0.2 | 0.2 $\pm$ 0.1 | 1.9 $\pm$ 1.1 |
| 22:6$\omega$-3 | 2.1 $\pm$ 1.5 | 5.6 $\pm$ 0.3 | 9.8 $\pm$ 3.2 |
| $\sum$**PUFA** | **28.5** | **48.9** | **42.7** |

Notes.
SFA, saturated fatty acids; MUFA, monounsaturated fatty acids; PUFA, polyunsaturated fatty acids.

**Table A.3** Mean fatty acid composition ($\pm$standard deviation) (percentage of total fatty acids detected) of mesozooplankton from the Chowan River by group.

| | Mesozooplankton groups | | | |
|---|---|---|---|---|
| | **G (1)** | **H (5)** | **I (5)** | **J (5)** |
| **14:0** | **4.5** | **5.4 $\pm$ 0.9** | **5.2 $\pm$ 1.1** | **5.1 $\pm$ 1.1** |
| 15:0 | 1.1 | 1.1 $\pm$ 0.4 | 0.7 $\pm$ 0.0 | 0.9 $\pm$ 0.1 |
| 16:0 | 29.7 | 24.4 $\pm$ 2.6 | 19.5 $\pm$ 1.3 | 22.0 $\pm$ 1.5 |
| 17:0 | 1.3 | 1.4 $\pm$ 0.4 | 1.0 $\pm$ 0.1 | 1.4 $\pm$ 0.3 |
| 18:0 | 11.2 | 7.7 $\pm$ 1.2 | 5.1 $\pm$ 0.3 | 6.8 $\pm$ 0.7 |
| 20:0 | 0.4 | 0.2 $\pm$ 0.0 | 0.2 $\pm$ 0.1 | 0.1 $\pm$ 0.1 |
| $\sum$**SFA** | **48.2** | **40.2** | **31.7** | **36.3** |
| 16:1$\omega$-9 | 0.3 | 0.9 $\pm$ 0.2 | 0.9 $\pm$ 0.4 | 0.8 $\pm$ 0.4 |
| 16:1$\omega$-7 | 4.1 | 9.5 $\pm$ 2.2 | 6.2 $\pm$ 2.2 | 8.6 $\pm$ 2.2 |
| 18:1$\omega$-9 | 17.0 | 10.6 $\pm$ 4.3 | 6.4 $\pm$ 1.2 | 4.0 $\pm$ 1.4 |
| 18:1$\omega$-7 | 2.1 | 3.9 $\pm$ 1.7 | 2.4 $\pm$ 0.4 | 3.0 $\pm$ 1.1 |
| 20:1 | 0.3 | 0.1 $\pm$ 0.1 | 0.3 $\pm$ 0.1 | 0.2 $\pm$ 0.1 |
| $\sum$**MUFA** | **23.8** | **25.0** | **16.2** | **16.6** |
| 18:2$\omega$-6 | 4.2 | 3.0 $\pm$ 0.9 | 5.1 $\pm$ 0.6 | 2.2 $\pm$ 0.7 |
| 18:3$\omega$-3 | 2.9 | 6.1 $\pm$ 2.1 | 10.7 $\pm$ 1.5 | 5.1 $\pm$ 1.9 |
| 18:4$\omega$-3 | 1.4 | 2.4 $\pm$ 0.8 | 7.1 $\pm$ 1.2 | 2.2 $\pm$ 0.4 |
| 20:2$\omega$-6 | 0.2 | 0.2 $\pm$ 0.1 | 0.4 $\pm$ 0.1 | 0.2 $\pm$ 0.0 |
| 20:3$\omega$-6 | 0.1 | 0.1 $\pm$ 0.0 | 0.1 $\pm$ 0.0 | 0.1 $\pm$ 0.0 |
| 20:4$\omega$-6 | 3.7 | 4.2 $\pm$ 1.5 | 2.1 $\pm$ 0.6 | 2.5 $\pm$ 1.2 |
| 20:3$\omega$-3 | 0.2 | 0.2 $\pm$ 0.1 | 0.5 $\pm$ 0.3 | 0.2 $\pm$ 0.1 |
| 20:4$\omega$-3 | 0.8 | 0.6 $\pm$ 0.3 | 1.2 $\pm$ 0.6 | 0.8 $\pm$ 0.3 |
| 20:5$\omega$-3 | 5.9 | 11.2 $\pm$ 2.4 | 13.8 $\pm$ 0.9 | 16.2 $\pm$ 0.6 |
| 22:5$\omega$-6 | 0.4 | 0.4 $\pm$ 0.2 | 0.8 $\pm$ 0.2 | 1.8 $\pm$ 0.8 |
| 22:5$\omega$-3 | 0.4 | 0.3 $\pm$ 0.2 | 0.4 $\pm$ 0.2 | 0.6 $\pm$ 0.3 |
| 22:6$\omega$-3 | 7.3 | 5.7 $\pm$ 2.9 | 10.1 $\pm$ 2.8 | 15.1 $\pm$ 7.3 |
| $\sum$**PUFA** | **27.5** | **34.4** | **52.3** | **47.0** |

Notes.
SFA, saturated fatty acids,; MUFA, monounsaturated fatty acids; PUFA, polyunsaturated fatty acids.

### Funding

This work was funded by NSF Grant DEB-0951414 to DG Kimmel. The funders had no role in study design, data collection and analysis, decision to publish, or preparation of the manuscript.

### Grant Disclosures

The following grant information was disclosed by the authors:
NSF: DEB-0951414.

### Competing Interests

The authors declare there are no competing interests.

### Author Contributions

- Deborah A. Lichti conceived and designed the experiments, performed the experiments, analyzed the data, contributed reagents/materials/analysis tools, wrote the paper, prepared figures and/or tables.
- Jacques Rinchard conceived and designed the experiments, performed the experiments, contributed reagents/materials/analysis tools, reviewed drafts of the paper.
- David G. Kimmel conceived and designed the experiments, analyzed the data, contributed reagents/materials/analysis tools, reviewed drafts of the paper.

### Data Availability

The raw data has been supplied as a Supplementary File.

### Supplemental Information

Supplemental information for this article can be found online at http://dx.doi.org/10.7717/peerj.3667#supplemental-information.

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
