# Peer review of "Changes in zooplankton community, and seston and zooplankton fatty acid profiles at the freshwater/saltwater interface of the Chowan River, North Carolina"

_PeerJ, doi:10.7717/peerj.3667_

## Round 0.1 · original submission · Major Revisions

Both reviewers indicated that a major revision of your ms is required and both pinpoint the data analyses. Please revise the data analyses based on your research questions and the objectives of the study, according to the recommendations of the referees. Furthermore, you should improve the clariry of the presentation for example by only discussing the main outcomes.

Reviewer 1 ·

Basic reporting

The ms provides valuable information on fatty acids (FAs) profiles of fish prey (zooplankton and seston) in an estuarine fish nursery in the Chowan River (North Carolina, USA). The study attempts to assess the variability of FAs from seston to zooplankton and to determine if these results can be used to indicate habitat quality for larval fish survival from a trophic viewpoint. Having said that, I see some problems in the thread of the work, the coherence between the objectives, the hypothesis, the experimental design and the treatment of the data to respond to the goal of the study. I believe that the work has potential but I feel thath it has not been exploited at all in the way it has been presented.

The ms is of an appropiate length and generally well-written but, although I am a non-native English speaker, I can see some grammatical errors and/or inconsistencies along it. Authors should check the manuscript carefully and correct the errors along it (some examples, line 44: organic “particles” instead of “participles”, line 46: through “direct” consumption instead of “direction”, line 97: remove one “that”, line 209: “among” river sections instead of “between”..you mentioned three river sections..). On the other hand, I think that the wording of the text of M&M and Results need to be improved to ensure greater clarity and understanding (an example of an unclear phrase, lines 211-216) .

References are sufficiently provided along the ms but in an unbalanced way. In some cases references are excesive (i.e. lines 329-331, 332-334, 374-376, 378-380...please use reviews) whereas in other cases they are absent and certainly needed (i.e. 43-45 definition of seston and references to substantiate this phrase??, 309-311 reference is required to substantiate that significant proportions of PUFA provide adequate nutrition for larval fish growth and development).

The general structure of the ms is appropiate but, again, M&M and Results should be reorganized and some parts, rewritten. Figures and table are understadable and readable.

I thank authors for providing the raw data, but I think it would be better if data had been presented sorted by sampling date and river sectors, that were the factors of the study.

The submission is self-contained but major improvements are required to be acceptable.

I recommend making some improvements along the ms. Please find them below.

Experimental design

I think the ms fits the scope of the journal. However, as I have noted above, it needs major improvements to be acceptable.

Authors sampled three planktonic components (seston, zooplankton and microzooplankton) in three sections of the river (upper, middle and lower sections) during three months of the year (April, May and June 2013). However, this experimental design was not considered at the time of establishing the objectives and analyzing the data accordingly.

I believe that the study requires the rethinking of the objectives and the analysis of the data according to the experimental design proposed.

Validity of the findings

Findings are interesting. However, the results must be validated by adding parametric statistics (transforming if necessary) and correcting errors in the application of multivariate statistics.
The discussion needs to be more concrete following the results without detailing them here. The figures should not be cited nor the groups generated in the analyzes. All that information should be avoided in the discussion. The discussion should be limited to analyzing the scope of the results. In this sense, it seems that the discussion needs to be polished and the information generated from this study needs certain decantation to be correctly analyzed within a theoretical context.

Additional comments

From my point of view, the two theoretical concepts that need to be carefully revised and correctely defined are “seston” and “habitat quality”. Seston refers to suspended particulate matter (SPM) not to organic compounds only (see Postel et al. 2000 In ICES Zooplankton Methodology Manual), so the phrase in lines 43-45 is not correct and this concept needs to be revised along the whole ms. The concept that seston or SPM, which includes all types of components and serves as food for zooplankton, may have a direct effect on zooplankton FA composition needs to be clearly explained. On the other hand, the quality of an habitat can be assessed in different manners, it is a very broad concept. I think this term needs to be defined in the context of the study. Are you referring to it as the trophic status to hold larval fish survival and development? Please defined it clearly and quotation marks will not be necessary at all (line 97).

I suggest to express the objectives more clearly following the experimental design authors proposed.
I would strongly recommend additional analysis using parametric statistic (ANOVAs) in order to compare FAs among planktonic components considering also the spatio-temporal factors authors have proposed. I also find problems in the multivariate analysis. The data of the present study have factors a priori defined, so ANOSIM can be used to test ONLY these factors NOT the groups defined by the cluster. Which is the hypothesis to be tested?? Please see Clarke & Warwick 2001: “A point that cannot be over-stressed is that ANOSIM only apply to groups of samples specified prior to seeing (or collecting) the data. A dangerous misconception is that one can use a cluster analysis of the species abundace data to define sample grouping, whose statistical validity can be established by perfoming an ANOSIM test for differences between groups. This is enterely erroneous, the argument being completely circular”...

To summarize, I would recommend authors make the effort to rethink the axis of the study, setting the objectives and the hypothesis according to the design approached. Please provide an analysis in accordance with the questions you have and present your results in an orderly manner. I think the work has potential but needs substantial improvements to be acceptable.

·

Basic reporting

The references are old and new ones need to be added. The figures and tables most are not necessary and a re-analysis needs to carried out with better analysis. Most of the comments are highlighted in the attached manuscript

Experimental design

Experimental design needs to be clearly explained so that the reader can easily follow the manuscript. they is confusion on what was actually done in the sampling section. See specific comments in the manuscript.

Validity of the findings

The study is sound but the data analysis carried out is wrong for the study. This could be greatly improved as suggested in the attachment

Additional comments

see attached manuscript and i did not go through the discussion as much as i would like to see an improved methods mainly data analysis and results section.

---

## Round 0.2 · Minor Revisions

I think the ms has been greatly imroved and is allmost ready for publication. The reviewer suggested a few very minor changes that you should attend to.

·

Basic reporting

The authors have addressed all the comments that we highlighted, the MS can be accepted for publication. The authors must be make small changes, its PERMANOVA not PermANOVA and also change the notation for FA 'n-' to 'ω', the commonly used notation.

Experimental design

This section has greatly improved and now reads well.

Validity of the findings

The aims, methods now support the results and all stats were necessary

Additional comments

The authors must be make small changes, its PERMANOVA not PermANOVA and also change the notation for FA 'n-' to 'ω', the commonly used notation.

---

## Round 0.3 · accepted · Accept

Your ms reads well and you made the required changes, so the ms is now ready for publication. Congratulations!